# Cooperative catalysis by a single-atom enzyme-metal complex

Xiaoyang Li[1,2,6], Yufei Cao[1,6], Kai Luo[3], Lin Zhang[4✉], Yunxiu Bai[1], Jiarong Xiong[1], Richard N. Zare [3✉] & Jun Ge [1,5✉]

Anchoring single metal atoms on enzymes has great potential to generate hybrid catalysts with high activity and selectivity for reactions that cannot be driven by traditional metal catalysts. Herein, we develop a photochemical method to construct a stable single-atom enzyme-metal complex by binding single metal atoms to the carbon radicals generated on an enzyme-polymer conjugate. The metal mass loading of Pd-anchored enzyme is up to 4.0% while maintaining the atomic dispersion of Pd. The cooperative catalysis between lipase-active site and single Pd atom accelerates alkyl-alkyl cross-coupling reaction between 1-bromohexane and $B$-n-hexyl-9-BBN with high efficiency (TOF is 540 h$^{-1}$), exceeding that of the traditional catalyst Pd(OAc)$_2$ by a factor of 300 under ambient conditions.

[1] Key Lab for Industrial Biocatalysis, Ministry of Education, Department of Chemical Engineering, Tsinghua University, 100084 Beijing, China. [2] State Key Laboratory of Food Science and Technology, School of Food Science and Technology, Nanchang University, 330047 Nanchang, China. [3] Department of Chemistry, Fudan University, Jiangwan Campus, 200438 Shanghai, China. [4] Department of Biochemical Engineering and Key Laboratory of Systems Bioengineering of the Ministry of Education, School of Chemical Engineering and Technology, Tianjin University, 300350 Tianjin, China. [5] Institute of Biopharmaceutical and Health Engineering, Tsinghua Shenzhen International Graduate School, 518055 Shenzhen, China. [6]These authors contributed equally: Xiaoyang Li, Yufei Cao. ✉email: linzhang@tju.edu.cn; zare@stanford.edu; junge@mail.tsinghua.edu.cn

Atomically dispersed metal catalysts have been emerging as an active frontier in heterogeneous catalysis due to the maximum atom efficiency and unique catalytic properties[1–5]. The construction of single-atom catalysts (SACs) with high activity, selectivity and stability is highly desired but remains a challenging goal. Anchoring single metal atoms on enzymes that theoretically merges enzymatic, homogeneous, and heterogeneous catalysis provide a promising platform for developing hybrid catalysts that can operate efficiently under ambient conditions. With the capability of recognizing and binding of reaction intermediates by enzyme active pocket, the activation energy of metal-catalyzed reactions could be largely reduced, and the efficiency and selectivity of metal catalysis could be greatly enhanced. The metal single atom-anchored enzyme could be an ideal catalyst for reactions that cannot be easily catalyzed by traditional metal catalysts. However, the combination of enzyme with metal single-atom catalysts has not been achieved yet, possibly because the flexible configuration and poor binding capability of protein cannot stabilize metal single atoms.

To address this challenge, in this work, we design a covalently bound enzyme–polymer conjugate and develop a photochemical method to produce carbon radicals on the polymer chain that bind to the metal single atoms, together with the amino acids on enzyme surface. The semiheterogeneous SACs constructed by anchoring single palladium (Pd) atoms on *Candida antarctic* lipase B (CALB) drives alkyl–alkyl cross-coupling reactions with high efficiency. The synthesis of alkyl–alkyl bonds[6–11] is one of the most important bond constructions in organic synthesis with numerous applications in pharmaceutical, materials, and energy industry. The alkyl–alkyl cross-coupling reactions cannot be catalyzed by original native lipase, while can be catalyzed by traditional Pd catalyst only at very low efficiency at ambient conditions. Herein, the turnover frequency of the lipase active site-adjacent Pd atom is 300 times higher than that of traditional catalyst $Pd(OAc)_2$ for alkyl–alkyl cross-coupling reaction between 1-bromohexane and *B*-n-hexyl-9-BBN in aqueous at 25 °C. It is found that the lipase active site serves as a scaffold to efficiently bind with alkyl chains, forming multiple noncovalent interactions, stabilizing the transition state of Pd-alkyl chain, and lowers the activation energy. Moreover, the selective binding of enzyme provides a selectivity for Pd catalysis.

## Results and discussion

**Photochemical synthesis of Pd-anchored enzyme**. Different from rigid carriers[12–15] normally used for the preparation of single-atom catalysts, protein molecules with flexible configuration and poor atom-binding capability cannot readily stabilize metal single atoms. Atomic dispersion of Pd on the enzyme–polymer conjugate was achieved by a photochemical method (Fig. 1a). The enzyme–polymer conjugate was synthesized by covalently linking enzyme with an aldehyde-functionalized triblock copolymer of propylene oxide and ethylene oxide (Pluronic F-127)[16]. After 20 min of the UV treatment of solution containing enzyme–pluronic conjugate, benzophenone (BP), and $PdCl_2$, the Pd atoms were captured on the enzyme–pluronic conjugate. By tuning the amount of $PdCl_2$ in the synthesis of Pd-anchored lipase–pluronic conjugate ($Pd_1$/CALB-P), the mass loadings of Pd can be controlled as high as 4.0% (w/w), with 16 Pd single atoms anchored on each CALB-P conjugate.

The sole presence of Pd atoms was confirmed by extended X-ray absorption fine structure spectroscopy (EXAFS). As shown in Fig. 1b and Supplementary Table 1, there was a notable peak centered at 1.57 Å from the coordination of Pd to O or C, and no Pd–Pd bond was detected. No obvious Pd clusters or nanoparticles were observed.

The image of aberration-corrected scanning transmission electron microscopy (AC-STEM) showed brighter dots, representing Pd atoms (Fig. 1c, Supplementary Fig. 1). Elemental mapping revealed the uniform distribution of Pd atoms (Fig. 1d and Supplementary Fig. 2). The Pd single atoms were also anchored on alcohol dehydrogenase (ADH), laccase (Lac), catalase (CAT), and glucose oxidase (GOx)–pluronic conjugates to demonstrate the generality of method (Supplementary Figs. 5 and 6). As controls, using enzyme or pluronic alone as the scaffold cannot obtain the atomically dispersed Pd (Supplementary Figs. 7 and 8).

The possible sites for anchoring Pd single atoms on enzyme were investigated. By density functional theory (DFT) calculations, the binding energies were calculated for six potential sites of enzyme–pluronic conjugate which can provide two O atoms to coordinate with $PdCl_2$, including Asp, Asn, Ser, Ser–Ser, Ser–Asp, and PEO chain (Supplementary Fig. 12). The Asp with negatively charged carboxyl group serves as an ideal ligand to form a stable four-coordinated structure with $PdCl_2$. It is the most likely site to anchor the Pd single atom with a binding energy of 2.53 eV (Supplementary Fig. 13). The possible anchoring positions of Pd single atoms were further investigated by nano-electrospray ionization mass spectrometry (nanoESI-MS). Three peptide fragments containing Pd single atoms were identified. The mass-to-charge ratio ($m/z$) of these fragments were 612 ($[(EMGGVVDNAAR) + Pd]^{2+}$), 625 ($[(SDAAR) + Pd]^+$), and 646 ($[(VLLSQNGTTPR) + Pd]^{2+}$) (Fig. 2a and Supplementary Fig. 14). These results also suggest that the Asp and Ser residues are most likely the binding sites for Pd single atoms.

In the photochemical synthesis of $Pd_1$/CALB-P, radicals were generated to react with $PdCl_2$ to stabilize the Pd single atoms on enzyme. The radical intermediates were trapped by 5,5-dimethyl-1-pyrroline N-oxide (DMPO) and characterized by electron spin resonance (ESR). The alkyl radical was identified in the photolysis of solutions containing both BP and CALB-P conjugate (Fig. 2d). In contrast, only ketyl radical (from BP) was detected in the solution containing BP and CALB without pluronic. This result reveals that photoexcited benzophenone readily abstracted H from the methylene group of pluronic (Fig. 2c and Supplementary Fig. 15). The alkyl radicals from pluronic then reacted with $PdCl_2$ that were first captured by amino acids of enzyme. The radical reaction removed $Cl^-$ from $PdCl_2$ as evidenced by EXAFS (Fig. 1b). The X-ray photoelectron spectroscopy (XPS) spectrum also indicated no existence of Cl in $Pd_1$/CALB-P (Supplementary Fig. 16). The high-resolution O 1*s*, C 1*s*, and Pd 3*d* XPS for the $Pd_1$/CALB-P indicated that both O–Pd and C–Pd bonds are present in the $Pd_1$/CALB-P, which is consistent with the EXAFS analysis (Fig. 2b, Supplementary Fig. 16). Based on the above results, the mechanism for the stabilization of Pd single atoms on enzyme is given in Fig. 2e, which consists of three steps: (1) the binding of $PdCl_2$ with two O atoms from the amino acid residues of enzyme to generate $PdCl_2$/CALB-P; (2) the formation of alkyl radicals via abstracting H from pluronic under UV irradiation; and (3) two alkyl radicals attack the Pd atoms, removing $Cl^-$ to produce $Pd_1$/CALB-P. The released $Cl^-$ ions were then captured by the ketyl radicals produced from BP (Supplementary Fig. 17). Without the photochemical process, Pd atoms cannot be stably anchored on the conjugate (Supplementary Figs. 18 and 19). The Pd-anchored enzyme preserved most of the protein structures (Supplementary Fig. 20 and Supplementary Table 6) and most of the enzymatic activities (Supplementary Fig. 21).

**Cooperative catalysis between lipase and Pd single atom**. The catalytic performance of $Pd_1$/CALB-P was evaluated in the alkyl–alkyl cross-coupling reactions in an aqueous solution at 25 °C. $Pd(OAc)_2$ was used as the reference catalyst[17]. The Pd

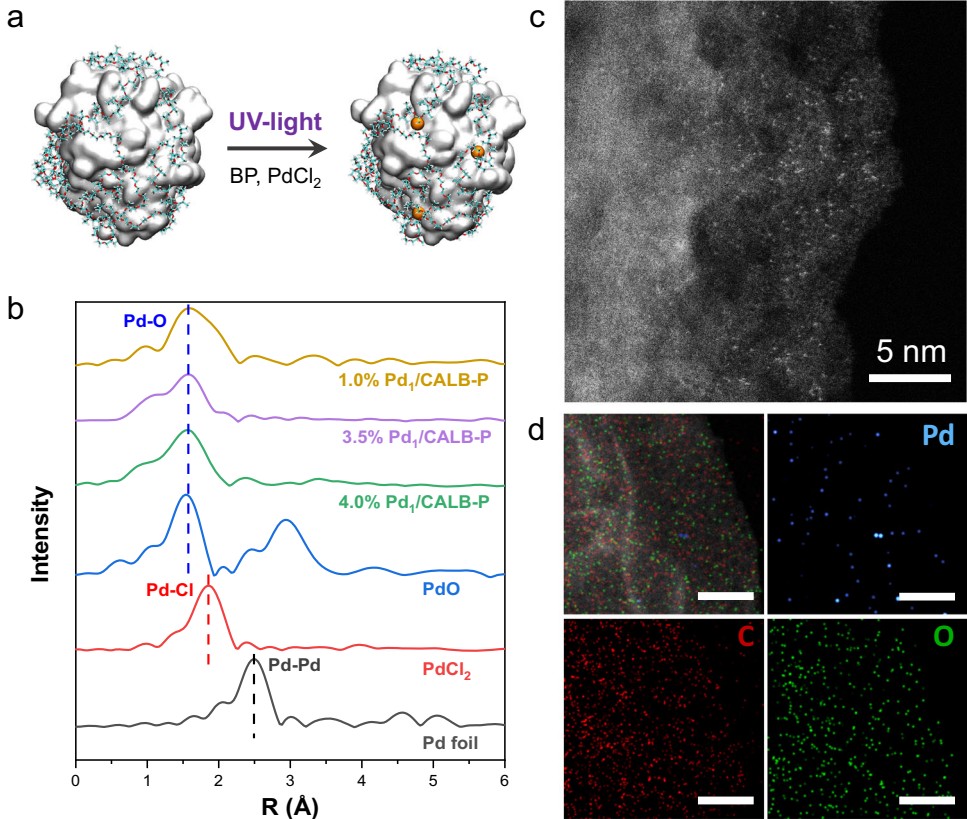

**Fig. 1 Preparation and characterizations of Pd1/CALB-P. a** Synthesis of Pd-anchored enzyme. **b** Pd K-edge EXAFS in R space for $Pd_1$/CALB-P. PdO, $PdCl_2$, and Pd foil were used as references. **c** Magnified AC-STEM image of $Pd_1$/CALB-P in which the bright dots represent Pd single atoms. **d** AC-STEM image of $Pd_1$/CALB-P and the corresponding elemental mapping for Pd, C, and O. Scale bars: 50 nm.

single atom was activated by employing trialkylphosphines as the electron-rich ligand to exchange with the O atom of amino acid for binding to Pd[8–11]. In the coupling of 1-bromohexane and B-n-hexyl-9-BBN, $Pd_1$/CALB-P showed a much higher activity than $Pd(OAc)_2$ (Fig. 3a). Calculating from the initial activity, the TOF for $Pd_1$/CALB-P ($33.8\,h^{-1}$) was 19-fold that of $Pd(OAc)_2$ ($1.8\,h^{-1}$) (Fig. 3b). In addition, $Pd_1$/CALB-P displayed at least an order of magnitude higher activity compared with a variety of other metal catalysts (Supplementary Table 7). Additionally, the $Pd_1$/CALB-P remained 75% of the activity after 10 catalytic cycles, indicating a good reusability (Fig. 3e).

The TOF value of $Pd_1$/CALB-P was decreased by 70% in the presence of pentadecane as the enzyme competitive inhibitor (Supplementary Fig. 23), demonstrating the important role of the active site of lipase in the Pd activity enhancement. When adding pentadecane at 5 min to occupy the enzyme active site of $Pd_1$/CALB-P, the rate of the alkyl–alkyl cross-coupling reaction was highly compromised (Fig. 3a and Supplementary Fig. 24). It was also found the activity enhancement of $Pd_1$/CALB-P generally disappeared (Fig. 3b) in THF, because the solvation of alkyl chain in THF reduced the binding of the alkyl halide in the active site of lipase[18]. The control samples of mixing $Pd(OAc)_2$ with CALB-P, CALB, or pluronic did not give obvious increase of Pd activity (Fig. 3b and Supplementary Fig. 22). Based on the analysis and theoretical calculations (Supplementary Figs. 12 and 13), we identified Asp223 is the most possible residue that anchors the Pd single atom on enzyme active site (Supplementary Fig. 25). It can be calculated that the TOF value of the enzyme active site-adjacent Pd atom is as high as $540\,h^{-1}$ (details can be found in Supplementary Fig. 24), which is 300-fold that of $Pd(OAc)_2$ (Fig. 3b).

The molecular dynamics (MD) simulation of the $Pd_1$/CALB-P–alkyl halide complex showed the distance of 1-bromohexane to the enzyme active site-adjacent Pd atom mainly distributed in 0.6–1.0 nm and the binding energy is in the range of −60 to −100 kJ/mol (Fig. 4b), indicating a high interaction between the artificial active site and the substrate. The reaction transition state formed between the enzyme active site, Pd atom and 1-bromohexane was identified with a binding energy of around −100 kJ/mol (Fig. 4a). In contrast, the distance of 1-bromohexane to the free Pd atom in solution has a wide distribution and the main distribution of binding energy is −1.0 to 1.0 kJ/mol (Fig. 4c). These results demonstrated that, similar to the mechanism of enzymatic catalysis, in the cooperative catalysis, when the active site of lipase binds the alkyl chain, the noncovalent interactions reduces the energy of the alkyl-Pd transition state and thus lowers the activation energy.

In a variety of cross-coupling reactions, $Pd_1$/CALB-P exhibited higher activity than $Pd(OAc)_2$ (Supplementary Fig. 26). It is noted that the longer the carbon chain of the alkyl halide, the higher activity of the Pd single atom on the enzyme (Fig. 3c). This is probably because the binding affinity of 1-bromododecane with the hydrophobic pocket of lipase is higher than that of 1-bromohexane (Fig. 3d, Supplementary Figs. 27, 28 and Table 8). Thus, the selective binding of enzyme provides a selectivity for Pd catalysis.

In summary, this study demonstrates a photochemical method for the construction of single-atom enzyme–metal complex. The selective recognition and binding of reaction transition state by enzyme active pocket significantly raise the activity and selectivity of the single atom catalysts. The combination of atomically dispersed metal catalysts with enzyme brings numerous

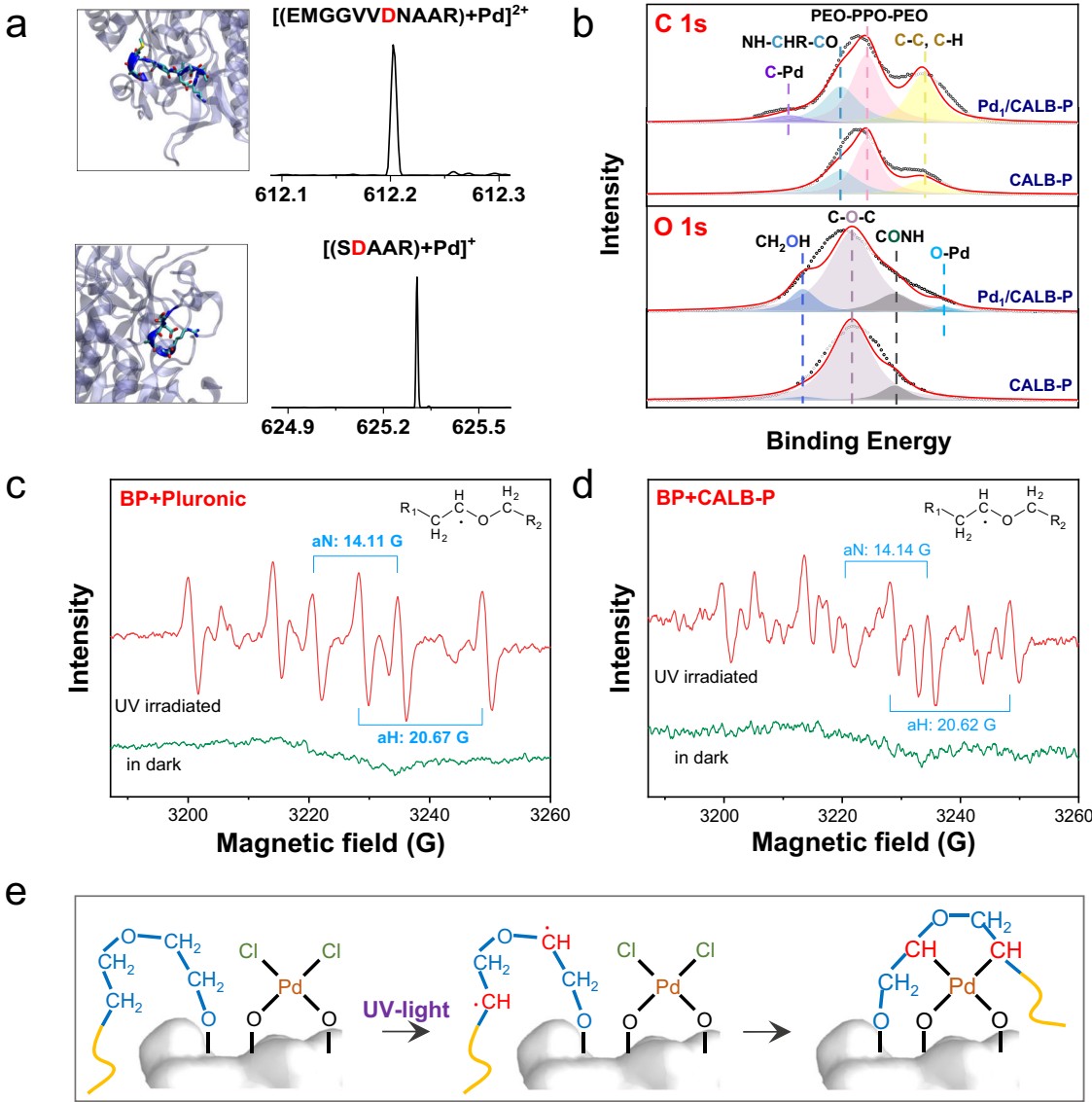

**Fig. 2 Mechanism for the stabilization of Pd single atoms on enzyme. a** The configurations and mass spectra of peptides of enzyme that possibly coordinated with Pd single atoms. **b** High-resolution C 1s and O 1s XPS spectra of Pd₁/CALB-P and CALB-P. ESR spectra of **c**, the mixture of BP and pluronic and **d**, the mixture of BP and CALB-pluronic after UV irradiation for 4 min or in dark in toluene. **e** Schematic illustration of the mechanism for the synthesis of Pd-anchored enzyme.

possibilities of designing catalysts to efficiently drive difficult to implement reactions at ambient conditions.

## Methods

**Chemicals**. Palladium chloride (PdCl₂, 99.9 wt%), benzophenone (99 wt%), *p*-nitrophenol (99 wt%) were purchased from Acros. *Candida Antarctic* lipase B (CALB, ≥5000 LU/G of liquid), glucose oxidase from *aspergillus niger* (GOx, 100,000–250,000 units/g solid), catalase from bovine liver (CAT, 2000–5000 units/mg protein), laccase from *agaricus bisporus* (Lac, ≥4 U/mg), alcohol dehydrogenase from *saccharomyces cerevisiae* (ADH, ≥300 units/mg protein), peroxidase from horseradish (HRP, ≥250 units/mg solid), 9-borabicyclo[3.3.1]nonane solution (9-BBN, 0.5 M in THF), nicotinamide adenine dinucleotide (NAD), 2,2′-azino-bis(3-ethylbenzothiazoline-6-sulfonic acid) (ABTS), Pluronic®F-127, *p*-nitrophenyl butyrate (*p*-NPB), ammonium acetate, 5,5-dimethyl-1-pyrroline N-oxide (DMPO) and palladium on activated carbon (Pd/C, extent of labeling: 10 wt% loading, matrix carbon), 1-nonene (96 wt%) were purchased from Sigma-Aldrich (St. Louis, MO, USA). Lipase was used after dialyzing against phosphate buffer (10 mM, pH 7.0) at 4 °C overnight. Dess-Martin periodinane, 1-butene (10% in hexane), 1-hexene (99 wt%), 1-octene (97 wt%), 1-bromohexane (99 wt%), 1-bromododecane (98 wt%), potassium phosphate monohydrate (K₃PO₄·H₂O) were purchased from Alfa Aesar. Hydrogen peroxide solution (30% H₂O₂), potassium carbonate (K₂CO₃), glucose, *N,N*-dimethylformamide (DMF),

methylene chloride, ethyl acetate, toluene, ethanol and ether were purchased from Sinopharm (China).

**Synthesis of single-atom enzyme–metal complex**. The enzyme–pluronic (enzyme–P) conjugates were fabricated using the method reported previously[16]. The terminal hydroxyl groups of Pluronic®F-127 (0.16 mmol) were first oxidized to aldehyde groups by using Dess–Martin periodinane (420.7 mg). The aldehyde-functionalized Pluronic®F-127 was subsequently bound to the enzymes via Schiff base reaction. The molar ratio of the amine groups in lysine of enzymes and the aldehyde groups in pluronic was 1:1.1. The powder of enzyme–P conjugates was obtained by lyophilization. To prepare the Pd₁/enzyme–P, Pd chloride (PdCl₂, 22.18 mg, 125 μmol) was first dissolved in dimethyl sulfoxide (5 mL) and enzyme–P (50 mg) was dispersed in dry toluene (5 mL). Benzophenone (30 mmol, 54.7 mg) was added to the enzyme–P solution under magnetic stirring at 25 °C. The mixture was then subjected to the removal of dissolved oxygen by bubbling argon for 30 min. To generate free radicals, the enzyme–P and BP mixture was irradiated under a xenon lamp with a UV filter (250–380 nm, 27.9 mW cm⁻², PLS-SXE300CUV). After 2 min of the UV irradiation, the PdCl₂ solution (100, 300, 500 μL) was added dropwise to the mixture for the preparation of 1.0%Pd₁/CALB-P, 3.5%Pd₁/CALB-P, 4.0%Pd₁/CALB-P, respectively. The volume of PdCl₂ solution was 100 μL in the preparation of 1.5% Pd₁/ADH-P, 0.5% Pd₁/Lac-P, 1.9% Pd₁/CAT-P, and 2.0% Pd₁/GOx-P. The UV irradiation continued for 18 min. The Pd₁/enzyme–P was collected by centrifugation at 4 °C, 11,100 × *g* for 5 min. After

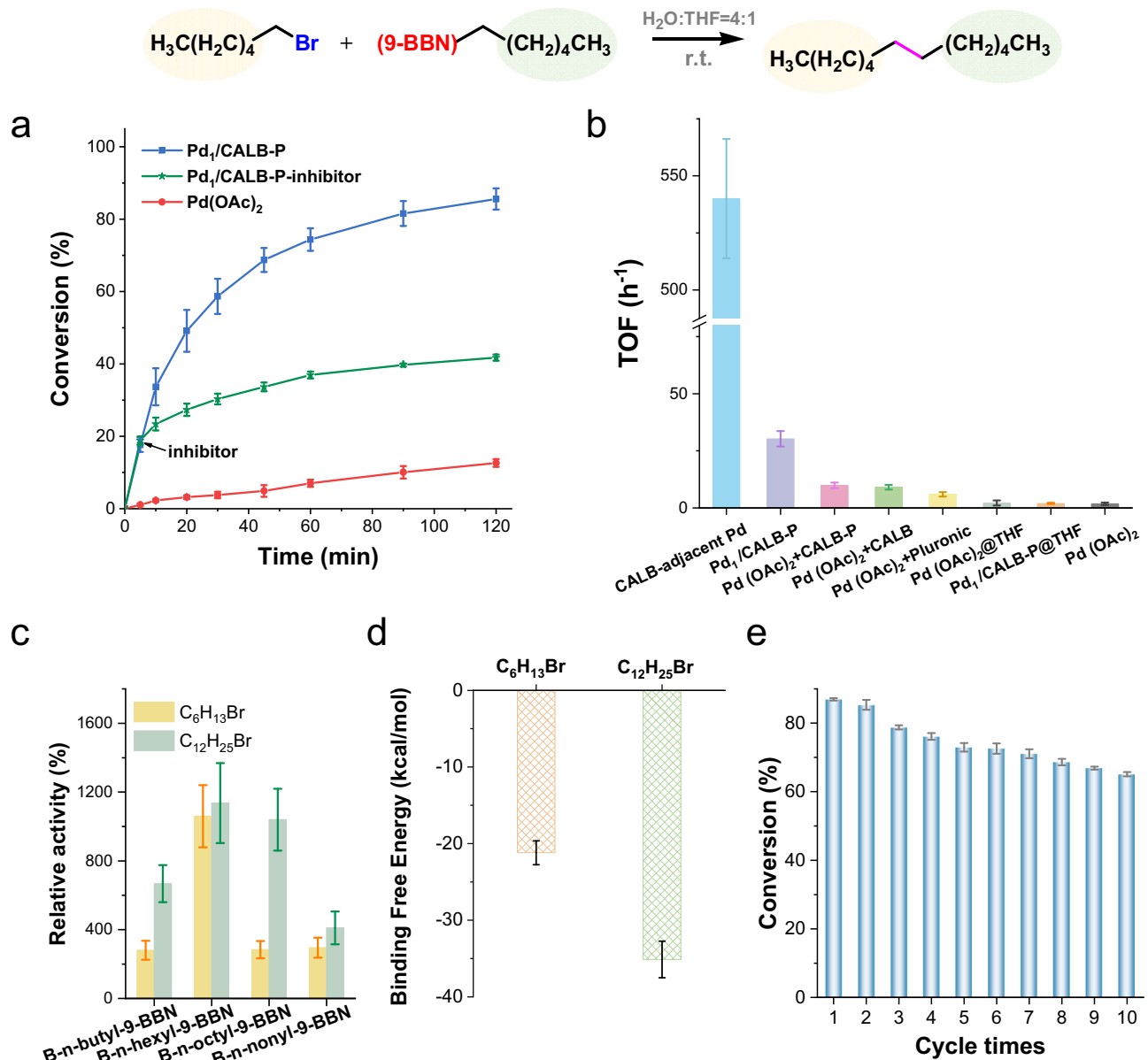

**Fig. 3 Catalytic performance of Pd1/CALB-P in alkyl–alkyl cross-coupling reactions. a** Reaction scheme and the conversion. **b** TOF of Pd1/CALB-P and Pd(OAc)2 in the cross-coupling of 1-bromohexane and *B*-n-hexyl-9-BBN in H2O:THF (4:1) or THF. Conditions: 30 mM 1-bromohexanes, 100 mM *B*-n-alkyl-9-BBN, and 6.7 mol% Pd. **c** Activity of Pd1/CALB-P in the cross-coupling reactions using 1-bromohexane or 1-bromododecane as the electrophile and *B*-n-butyl-9-BBN, *B*-n-butyl-9-BBN, *B*-n-butyl-9-BBN, or *B*-n-butyl-9-BBN as the nucleophile, compared with Pd(OAc)2 (reference activity as 100%) at the same amount of Pd. **d** The calculated binding free energies of the enzyme-substrate complex. **e** Conversions in ten cycles for the cross-coupling of 1-bromohexane and *B*-n-hexyl-9-BBN reaction using Pd1/CALB-P as the catalyst in H2O:THF (4:1). Each data point and error bar represent the mean and standard deviation from at least three independent measurements.

washing with toluene (5 mL × 3), the precipitated Pd1/enzyme–P was then washed by ethanol (5 mL × 3) and resuspended in deionized water (5 mL). The powder of the Pd1/enzyme–P was obtained by lyophilization and stored at 4 °C.

**Alkyl–alkyl cross-coupling catalyzed by Pd1/CALB-P**. To prepare *B*-n-alkyl-9-BBN, the terminal alkene (5.0 mmol) was added to a 25 mL vessel which was sealed with a septum and then was purged with argon for 10 min. The 9-BBN solution (5.0 mmol, 10 mL of a 0.50 M solution in THF) was added to the vessel by syringe. The mixture was then stirred vigorously at 25 °C for 16 h. The yield of *B*-n-alkyl-9-BBN was determined by GC–MS analysis. Pd1/CALB-pluronic (28 mg, 0.01 mmol Pd), PCy3 (56 mg, 0.2 mmol), K3PO4·H2O (345 mg, 1.50 mmol) and deionized water (4 mL) were added to a reaction vessel equipped with a stir bar. 1-Bromohexane (0.15 mmol, 21 μL) and *B*-n-alkyl-9-BBN solution (1 mL; 0.50 M solution in THF) were then added. The reaction mixture was stirred vigorously at 25 °C. The TOF values were calculated at the beginning of the reactions (at 5 min).

**Nano-electrospray ionization mass spectrometry (nano-ESI)**. The nanoESI source is composed of a pulled glass capillary with a 4 cm steel wire inserted. The applied voltage is 0.9–3.5 kV. The inner diameter and outer diameter of borosilicate glass capillary tips was 0.86 and 1.5 mm, respectively. The microdroplet at the tip of nanoESI was formed by applying high voltage on the needle. The concentration ratio of protein and trypsin (dissolved in 5 mM ammonium acetate solution at pH 6.0) in the nanoESI solution was 20:1. The final protein concentration was 5–20 μM. The ambient MS analysis was performed on LTQ Orbitrap Velos mass spectrometer (Thermo Scientific, San Jose, CA, USA). The MS capillary temperature was 275 °C with the S-lens voltage of 55 V.

**Molecular dynamics simulations**. All-atom (AA) model of lipase was constructed based on the crystal structure in Protein Data Bank (PDB ID: 1TCA, http://www.rcsb.org/pdb/)[19] and CHARMM36 force field. Pluronic composed of (EO)100−(PO)65−(EO)100 was used and the parameters were set according to the parameters of similar structure in CHARMM36 force field. An initial structure of

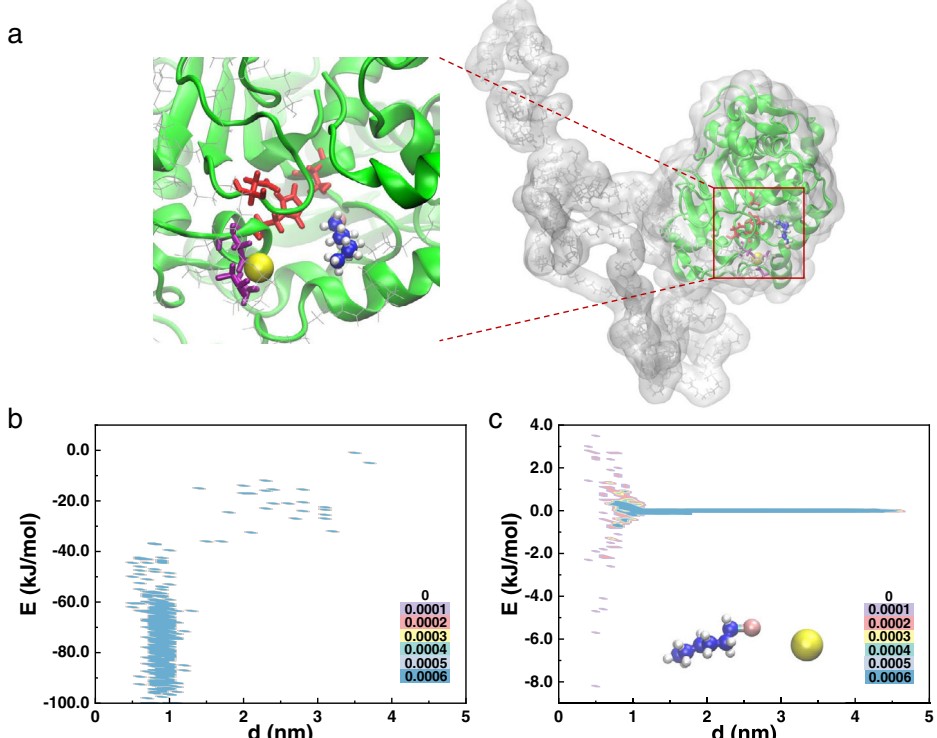

**Fig. 4 Molecular dynamics simulations of Pd1/CALB-P binding with 1-bromohexane and forming reaction transition state. a** All-atom model of $Pd_1$/CALB-P. The enlarged region of the representative snapshots gives the reaction transition state which is stabilized by the active site with a binding energy of 104.5 kJ/mol. Details can be found in supplementary materials. The distribution probability plotted as a function of interaction energy and distance for **b**, 1-bromohexane to the enzyme active site–adjacent Pd atom and **c**, 1-bromohexane to the free Pd atom in solution. The distribution probability is shown by the filling colors.

substrate with $Pd_1$/CALB was obtained using molecular docking. The $Pd_1$/CALB was used as the receptor, while the substrate was used as the ligand. AUTODOCKTOOLS 1.5.6.[20] was used for preparing the receptor and ligand. All polar hydrogens were added and the Kollman United Atomic charges were computed in the receptor. The Gasteiger charges were computed in the ligand. Grid definition was set up following the recommendations of the program manual[20]. The ligand was docked to collagen by using AUTODOCK VINA 1.1.2[21] (http://vina.scripps.edu/). The Pd atom was replaced by Zn atom due to the lack of forcefield parameters in the docking. The affinity energy value (kcal/mol) for each ligand was calculated and denoted by $E_{VINA}$. The best docked conformation was chosen, and pluronic was subsequently linked via Lys 136 of lipase to get the initial structure of substrate with $Pd_1$/CALB-P for the following MD simulations. The initial conformation of the substrate with $Pd_1$/CALB-P was placed in the center of a cubic box ($12 \times 13 \times 10$ nm)[21]. For the system of free Pd, a complex of Pd and a substrate was extracted from the system of $Pd_1$/CALB-P and then placed in the center of a cubic box ($6 \times 6 \times 6$ nm)[21]. MD simulations in the NVT ensemble were performed by GROMACS 5.1.4[22, 23]. Temperature was controlled at 298.15 K by the velocity-rescale (v-rescale) method[24]. The time constant was 0.5 ps. linear constraint solver (LINCS) algorithm[25] was applied to constrain all bonds. The electrostatic interaction was calculated by using particle-mesh Ewald (PME) algorithm[26]. The Lennard–Jones (LJ) potential, Coulomb potential energies, and cutoffs of neighbor atom list were all set as 1.8 nm. The initial velocities of particles were generated according to Maxwell distribution. The Verlet algorithm was used for integration with a time step of 2 fs. An energy minimization was performed, followed by 100 ns MD simulation. Four independent simulations were performed for each set of condition. The gmx energy program in GROMACS was used for the calculations of the potential energies between the substrate and $Pd_1$/CALB-P. The total binding energy (denoted as E) was calculated as the sum of LJ and Coulomb potential energies. The gmx mindist program in GROMACS was used for the calculations of the minimum distance between the substrate and $Pd_1$/CALB-P or free Pd atom (denoted as d). The distribution probability (denoted by P) was defined and calculated as the percentage of duration time with certain E and d during the simulation. The VMD software (http://www.ks.uiuc.edu/Research/vmd/) and Rasmol program were used for the preparation of snapshots.

**Binging energy of potential surface sites of enzyme–polymer conjugate.** The abundant oxygen atoms on the surface of enzyme–polymer conjugate provide suitable anchor sites for $PdCl_2$ to form a stable four coordination structure. Two adjacent oxygen atoms are needed to bind with one Pd atom. Through the analysis of the oxygen atoms on the surface, six potential types of oxygen atom pairs were found as shown in

Supplementary Fig. 12. They are two carboxyl oxygen atoms on the side chain of Asp, two carbonyl oxygen atoms on Asn, hydroxyl and carbonyl oxygen atoms on Ser, two hydroxyl oxygen atoms on two adjacent Ser (named Ser–Ser), hydroxyl and carboxyl oxygen atoms on adjacent Ser and Asp, and two adjacent oxygen atoms on PEO chain, respectively. When the oxygen atom pairs are from different amino acids, in order to ensure the original position of the two amino acids on the protein, we fixed the Cα and adjacent C, N of the amino acid during the optimization of the geometric conformation in DFT calculation. The confirmation of enzyme–polymer conjugate in methylbenzene was obtained from our previous work[16].

**Preparation of Pd/Pluronic.** The aldehyde-functionalized Pluronic®F-127 (50 mg) was dissolved in dry toluene (5 mL). Benzophenone (30 mmol, 54.7 mg) was added and the mixture was then subjected to the removal of dissolved oxygen by bubbling argon for 30 min. Palladium chloride (22.18 mg, 125 μmol) was dissolved in dimethyl sulfoxide (5 mL). To generate free radicals, the Pluronic and BP mixture was irradiated under a UV lamp. After 2 min of the UV irradiation, the $PdCl_2$ solution (100 μL) was added dropwisely to the mixture. The UV irradiation was continued for 20 min. The product was collected by centrifugation at 4 °C, 11,100×g for 5 min. After washing with toluene (5 mL × 3), the precipitated Pd/Pluronic was then washed by ethanol (5 mL × 3) and resuspended in deionized water (5 mL). The powder of the Pd/Pluronic was obtained by lyophilization and stored at 4 °C.

**Preparation of Pd/CALB.** CALB (50 mg) was dissolved in dry toluene (5 mL). Benzophenone (30 mmol, 54.7 mg) was added and the mixture was then subjected to the removal of dissolved oxygen by bubbling argon for 30 min. Palladium chloride (22.18 mg, 125 μmol) was dissolved in dimethyl sulfoxide (5 mL). To generate free radicals, the CALB and BP mixture was irradiated under a UV lamp. After 2 min of the UV irradiation, the $PdCl_2$ solution (100 μL) was added dropwisely to the mixture. The UV irradiation was continued for 20 min. The product was collected by centrifugation at 4 °C, 11,100 × g for 5 min. After washing with toluene (5 mL × 3), the precipitated Pd/CALB was then washed by ethanol (5 mL × 3) and resuspended in deionized water (5 mL). The powder of the Pd/CALB was obtained by lyophilization and stored at 4 °C.

**Preparation of PdCl2/CALB-P.** The CALB-P conjugates (50 mg) were dispersed in dry toluene (5 mL). Palladium chloride (22.18 mg, 125 μmol) was dissolved in dimethyl sulfoxide (5 mL) and then 500 μL was added dropwisely to the mixture

under magnetic stirring at 25 °C. After reacting for 20 min, the product was collected by centrifugation at 4 °C, 11,100 × g for 5 min. The obtained sample was denoted as $PdCl_2$/CALB-P. After washing with toluene (5 mL × 3), the precipitate was then washed by ethanol (5 mL × 3) and resuspended in deionized water (5 mL). The powder of the $PdCl_2$/CALB-P was obtained by lyophilization and stored at 4 °C.

**Enzyme activity assays of single-atom enzyme–metal complex**. The activities of CALB and $Pd_1$/CALB-P were assayed by a standard method using p-NPB as the substrate. The activities of ADH and $Pd_1$/ADH-P were assayed by a standard method using ethanol and nicotinamide adenine dinucleotide as the substrates. The activities of Lac and $Pd_1$/Lac-P were assayed by a standard method using ABTS as the substrate. The activities of CAT and $Pd_1$/CAT-P were assayed by a standard method using hydrogen peroxide as the substrate. The activities of GOx and $Pd_1$/GOx-P were assayed by a standard method using glucose as the substrate.

**Characterization**. The X-ray absorption data at the Pd K-edge of the single-atom enzyme–metal complexes were recorded at the XAFS station BL14W1 of the Shanghai Synchrotron Radiation Facility (SSRF), China. ESR measurements were performed on a JEOL FA-200 apparatus, with 9057 MHz microwave frequency and 4.00 mW power. The ESR measurements of samples after UV irradiated were proceed by in situ UV irradiation under a Xenon lamp with the wavelength of 365 nm. The Pd loadings on catalysts were measured with an ICP-OES on an IRIS Intrepid II XSP instrument (Thermo Electron Corporation). AC-STEM images were acquired using a FEI Titan 80-300 transmission electron microscopy operated at 300 kV. Before the aberration-corrected STEM characterization, $Pd_1$/CALB-P was adsorbed on GO and then calcined for better contrast. In a typical process, $Pd_1$/CALB-P (2 mg) was dispersed in deionized water (1 mL) and mixed with GO (50 mg) in an alumina crucible. After dried in vacuum oven, the mixture was then placed into a muffle furnace with a controlled temperature program. The temperature was increased by 1 °C/min from room temperature to 250 °C and allowed to stay at 250 °C for 30 min. The calcination process was carried out in air. HAADF-STEM and HRTEM were recorded on a JEOL JEM-2100F instrument operated at 200 kV. TEM characterization was conducted on a JEOL JEM-2010 instrument working at 120 kV. XPS analysis was performed on a Thermal Scientific ESCALAB 250Xi instrument using an monochromatized Al Kα X-ray source (E = 1486.6 eV) operating at 15 kV and 15 mA. A GC system (Agilent 7890B) and a triple quadrupole mass spectrometer (Agilent 7000D) were used for the GC–MS analysis. The GC was equipped with a HP-5 MS capillary column (30 m × 0.25 mm) (Agilent) operating at an initial temperature of 60 °C with helium carrier gas (column flow rate = 1.0 mL min$^{-1}$). The activities of enzymes were determined on a UV/Vis spectrophotometer SHIMADZU UV-2600. The apparent TOF of Pd catalysts was calculated by dividing the rate at the beginning of cross-coupling reactions by the number of surface accessible Pd atoms, which can be expressed in mol of product formed/(surface mol of Pd × h).

## Data availability
The data that support the findings of this study are available from the corresponding author upon request.

## Code availability
The code used in this study is available from the corresponding author upon request.

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

## Acknowledgements
This work was supported by the National Key Research and Development Plan of China (2021YFC2102800), the National Natural Science Foundation of China (21878174, 22168024), and the Tsinghua-Foshan Innovation Special Fund (TFISF). We thank beamline BL14W1 (Shanghai Synchrotron Radiation Facility) for providing the beam time.

## Author contributions
J.G. and X.L. conceived the idea. X.L. performed the experiments with technical help from Y.B. and J.X. Z.L. and Y.C. performed the calculations. K.L. and R.N.Z. performed the mass spectra analyses. X.L., Z.L., Y.C., J.G., and R.N.Z. co-wrote the paper.

## Competing interests
The authors declare no competing interests.
