## [Peer Review File · Nature Communications]

Editorial Note: This manuscript has been previously reviewed at another journal that is not operating a transparent peer review scheme. This document only contains reviewer comments and rebuttal letters for versions considered at *Nature Communications* .

REVIEWERS' COMMENTS

Reviewer #2 (Remarks to the Author):

The manuscript is improved and I can support its publication in Nat Commun. I'm still not 100% sold on the general applicability or even practical utility but I'm willing to give the authors the benefit of the doubt.

Reviewer #3 (Remarks to the Author):

The authors reported a photochemical method to construct Pd-single atom-anchored enzymes. After characterizations and calculations, the authors proposed that the possible anchoring positions of Pd single atoms are Asp and Ser residues. Interestingly, the authors demonstrated that the catalytic ability of such single atom-anchored enzymes in an alkyl-alkyl cross-coupling reactions was evident improved comparing to Pd(OAc)₂. The discovery that 1-bromododecane has different reactivity than 1-bromohexane is also interesting, highlighting the potential advantages of these artificial systems. Overall, this reviewer thinks that the development of such a platform for hybrid catalysis could lead to new reactivities and the contents are suitable for publishing in Nat. Comm.

Responses to reviewers' comments

Reviewer #2:

Comment 1. The manuscript is improved and I can support its publication in Nat Commun. I'm still not 100% sold on the general applicability or even practical utility but I'm willing to give the authors the benefit of the doubt.

Response: Thank you very much for the positive assessment.

Reviewer #3

Comment 1. The authors reported a photochemical method to construct Pd-single atom-anchored enzymes. After characterizations and calculations, the authors proposed that the possible anchoring positions of Pd single atoms are Asp and Ser residues. Interestingly, the authors demonstrated that the catalytic ability of such single atom-anchored enzymes in an alkyl-alkyl cross-coupling reactions was evident improved comparing to Pd(OAc)₂. The discovery that 1-bromododecane has different reactivity than 1-bromohexane is also interesting, highlighting the potential advantages of these artificial systems. Overall, this reviewer thinks that the development of such a platform for hybrid catalysis could lead to new reactivities and the contents are suitable for publishing in Nat. Comm.

Response: Thank you very much for the positive assessment.